# Bone Regeneration of Critical-Size Calvarial Defects in Rats Using Highly Pressed Nano-Apatite/Collagen Composites

**DOI:** 10.3390/ma15093376

**Published:** 2022-05-08

**Authors:** Wataru Hatakeyama, Masayuki Taira, Tomofumi Sawada, Miki Hoshi, Yuki Hachinohe, Hirotaka Sato, Kyoko Takafuji, Hidemichi Kihara, Shinji Takemoto, Hisatomo Kondo

**Affiliations:** 1Department of Prosthodontics and Oral Implantology, School of Dentistry, Iwate Medical University, 19-1 Uchimaru, Morioka 020-8505, Iwate, Japan; whatake@iwate-med.ac.jp (W.H.); hoshmiki@iwate-med.ac.jp (M.H.); ykhcnh@iwate-med.ac.jp (Y.H.); takafuji@iwate-med.ac.jp (K.T.); hkihara@iwate-med.ac.jp (H.K.); hkondo@iwate-med.ac.jp (H.K.); 2Department of Biomedical Engineering, Iwate Medical University, 1-1-1 Idaidori, Yahaba-cho, Shiwa-gun 028-3694, Iwate, Japan; sawada@iwate-med.ac.jp (T.S.); takemoto@iwate-med.ac.jp (S.T.); 3Division of Anatomical and Cellular Pathology, Department of Pathology, Iwate Medical University, 1-1-1 Idaidori, Yahaba-cho, Shiwa-gun 028-3694, Iwate, Japan; staisei@iwate-med.ac.jp

**Keywords:** nano-hydroxyapatite, collagen composite, hydraulic press, bone regeneration, osteo-conduction, micro-computed tomography, bone remodeling

## Abstract

Osteo-conductive bone substitute materials are required in dentistry. In this study, highly pressed nano-hydroxyapatite/collagen (P-nHAP/COL) composites were formed by a hydraulic press. Critical-size bone defects (Φ = 6 mm) were made in the cranial bones of 10-week-old Wistar rats, in which P-nHAP/COL and pressed collagen (P-COL) specimens were implanted. Defect-only samples (DEF) were also prepared. After the rats had been nourished for 3 days, 4 weeks, or 8 weeks, ossification of the cranial defects of the rats was evaluated by micro-computed tomography (micro-CT) (*n* = 6 each). Animals were sacrificed at 8 weeks, followed by histological examination. On micro-CT, the opacity of the defect significantly increased with time after P-nHAP/COL implantation (between 3 days and 8 weeks, *p* < 0.05) due to active bone regeneration. In contrast, with P-COL and DEF, the opacity increased only slightly with time after implantation, indicating sluggish bone regeneration. Histological inspections of the defect zone implanted with P-nHAP/COL indicated the adherence of multinucleated giant cells (osteoclasts) to the implant with phagocytosis and fragmentation of P-nHAP/COL, whereas active bone formation occurred nearby. Fluorescent double staining indicated dynamic bone-formation activities. P-nHAP/COL is strongly osteo-conductive and could serve as a useful novel bone substitute material for future dental implant treatments.

## 1. Introduction

To restore highly atrophic alveolar ridges, alloplasts (bone substitute material) are frequently applied in dental implant therapy [1]. Although autogenous bone grafts are considered the gold standard [2], alloplasts are regarded as safer and more patient-friendly (i.e., less invasive) materials [3]. Further development of alloplasts has the potential to remarkably improve the current success rate of implant treatments (e.g., by enhancing vertical bone augmentation [3]).

Calcium phosphates, such as hydroxyapatite (HAP) and tricalcium phosphate, have been used as bone substitute materials in various configurations, including particulates, plates, and blocks [4,5]. These large-sized materials show high biocompatibility and slow osteo-conductivity, although HAP has a slower degradation rate than tricalcium phosphate in vivo [5]. Furthermore, nano-hydroxyapatite (nHAP) has recently received considerable attention in the biomaterials research community because of specific characteristics such as superior bio-absorbability, osteo-conduction, and osteo-induction compared with conventional macro- and micro-sized HAP [6,7,8]. On the one hand, a practical disadvantage of nHAP is the difficulty associated with its handling; generally, nHAP is powdery and easily phagocytized by macrophage in vivo [9]. To overcome this disadvantage, nHAP powder has been combined with collagen (COL) as a scaffold material (a binder), thereby yielding nHAP/COL composite [10]. In our previous in vivo study, nHAP/COL composite (soft, porous sponge without pressing) was prepared by mechanical mixing and freeze drying [11]. This scaffold material indicated osteo-conductive effects at 4 weeks in rat calvarial bone defects; however, new bone formation at 8 weeks was slightly decreased due to the rapid bio-absorption of the material.

Thus, further improvement was needed for achieving a prolonged osteo-conductive effect of nHAP/COL composite. One possible method to address this requirement is the condensation of the composite using a hydraulic press [12]. This technique has already been employed to decellularize and sterilize animal skin and meat in the food industry [13,14]. Under pressing, the collagen structure (e.g., α-helix and β-sheet) is significantly altered, changing the functional and structural properties of collagen, dependent on the magnitude and period of pressure [13,14]. Pressed collagen might possess increased hydrophobicity, compressive strength, and in vivo longevity [13]. On the other hand, for preparation of HAP, cold isostatic pressing was used as one process for raw materials prior to sintering [15]. Although once sintered HAP might not be chemically altered by pressing, pressing might cause HAP particles to agglomerate, leading to different physical properties [16]. In addition, hot isostatic pressing has often been employed to produce dense and highly crystalline HAP [17]. In our previous studies, the pressing technique could embed HAP particles within COL with greater physical energy and the osteogenic differentiation of osteoblasts was accelerated on pressed nHAP/COL (P-nHAP/COL) composite compared with pressed COL(P-COL) in vitro [12,18]. However, the evidence of P-nHAP/COL is still insufficient for clinical use.

Up to now, there have been little reports concerning the effect of high pressure on not only physical and chemical properties but also in vivo osteo-conduction of P-nHAP/COL composites. It is highly expected to unveil these properties. Currently, most bulk scaffold materials are porous with interconnected porosity [19]. P-nHAP/COL composites are, however, plain and dense without pore structure, and their usefulness remains unknown. The novelty of this research lies in their clarification.

A rat-skull critical-size defect model has been employed in many previous studies to evaluate bone regeneration by osteo-conductive materials [20,21]. Three-dimensional (3D) micro-computed tomography (micro-CT) analysis is commonly used to observe microscopic bone structures and quantify bone formation within bony defects [22,23]. Histological observations often follow, using non-decalcified sliced specimens stained with Villanueva solution and fluorescent double labeling [24].

Therefore, in this study, P-nHAP/COL composites were prepared by mechanical mixing, freeze drying, and hydraulic press; these were then evaluated to determine their utility as osteo-conductive bone substitute materials in a rat-skull critical-size bone-defect model using micro-CT analyses and histological observations.

## 2. Materials and Methods

### 2.1. Preparation of P-nHAP/Col Composite Material

Virus-free medical grade COL pellets (NMP collagen PS, Nippon Meat Packers Inc., Tokyo, Japan) (1 g) were dissolved in distilled water (28 mL) in 50-mL polystyrene conical tubes (Greiner Bio-one, Frickenhausen, Germany) at 4 °C. The resulting acidic solution was neutralized with a 0.1 N NaOH solution (6.5 mL) in a plastic dish (100 × 70 × 12 mm) to obtain a COL gel with an appropriate pH (7.5). A mixture of COL gel and nHAP powder in the solution (MHS-00405 type nano SHAp, Sofsera, Tokyo, Japan) (1.5 g) was prepared by manual mixing. This nHAP powder material was spherical in shape and exhibited a very small average particle size (40 nm) [25]. The mixture gel was frozen at −80 °C for 3 h and freeze dried (FD-5N, Eyela, Tokyo, Japan) overnight. The resulting sponge was successively cross-linked by dehydrothermal treatment at 140 °C for 24 h in a vacuum-dry oven (VO-300, AS ONE, Tokyo, Japan). A hydraulic press (NT-100H, Sansho Industry, Osaka, Japan) (Figure 1a) ensured that the material in the metal mold (9.95 mm in inner diameter and 20 mm in height) was consistently pressurized, while manual and oil-produced pressure (29.4 kN; 3000 kgf) were applied concurrently from the top and bottom (i.e., co-axial bidirectional pressing) for 2 min. The resulting sheet, prepared by high pressure (P-nHAP/COL) (Figure 1b), was punched out to obtain specimens approximately 6 mm in diameter and 1 mm in height. As controls, punched pressed collagen (P-COL) specimens without nHAP were used. These disks were sterilized with ethylene oxide gas for at least 24 h and kept in a vacuum desiccator.

Scanning electron microscopy (SEM) observations of both COL and nHAP/COL before and after hydraulic press were performed by using a scanning electron microscope (SU8010, Hitachi High-Tech Corp., Tokyo, Japan) at 2 or 10 kV after plasma coating with OsO_4_ to understand morphological changes by hydraulic press. In addition, the pore sizes of COL and nHAP/COL sponges (pre-forms of P-COL and P-nHAP/COL) were calculated from different locations (*n* = 20) per one sample.

### 2.2. Animal Experiments

Eighteen male Wistar rats weighing 340 ± 16 g (mean ± SD) were used. All rats were housed in separate cages (three rats per cage) with standard diet and water ad libitum. Under anesthesia with a mixture of isoflurane (3 vol.%) and oxygen (0.5 L/min) gas generated by a carburetor (IV-ANE, Olympus, Tokyo, Japan), the centers of the rat calvariae were shaved and sterilized with 10% povidone iodine, followed by local injection of anesthetic (0.2 mL, 2% lidocaine with 1:80,000 epinephrine). Then, full-thickness periosteum flaps were elevated and bone defects were created using a trephine bur (6 mm in diameter; Implant Re Drill System, GC, Tokyo, Japan). Six P-nHAP/COL and six P-COL specimens were implanted in rat calvarial bone defects, and six holes were left empty (DEF). Each flap was repositioned and closed with soft nylon (Softretch 4-0, GC). At 8 weeks after surgery, all rats were sacrificed by CO_2_ inhalation. Animal experiments were performed in accordance with the guidelines for the care and use of laboratory animals and approved by the Institutional Ethics Committee of Iwate Medical University on 9 September 2013 (approval number: #25-015).

### 2.3. Micro-CT Imaging

New bone formation in the defect area of rat calvarial bone that contained P-nHAP/COL or P-COL was examined by using a 3D micro-CT system (eXplore Locus, GE Healthcare, Boston, MA, USA). All samples were scanned at 90 μm intervals at 80 kV and 450 μA, and Vextus Factor-compiled storage files (VFF data) were acquired. After scanning, transverse reformatted micro-CT images of the calvariae were reconstructed using 3D imaging analysis software (MicroView Version 2.2, GE Healthcare). The instrumental opacity threshold value was set to 8000 to minimize the interference from other bony tissues (e.g., maxillary bones and the cranial base). ImageJ 1.53k software (National Institutes of Health, Bethesda, MD, USA) was used to quantify the radiographic opacity of the defect area containing P-nHAP/COL or P-COL. DEF was also evaluated.

Micro-CT measurements (i.e., X-ray opacity) were statistically assessed with respect to intra- and inter-individual differences via repeated-measures analysis of variance (ANOVA) followed by the least significant difference post-hoc test, using SPSS software (version 16.01; SPSS Inc., Chicago, IL, USA). Intra- and inter-individual differences were considered significant if p-values were less than 0.05 (*p* < 0.05), and highly significant if *p*-values were less than 0.01 (*p* < 0.01). Tukey multiple comparison tests were used to determine which pairs of groups were significantly different. Graphs were generated using Kaleida graph software (version 4.5.3, Hulinks Inc., Tokyo, Japan).

### 2.4. Histological Observations

Fluorescent double staining was performed on one of six rats implanted with P-nHAP/COL at 8 weeks of age. Sequential labeling was performed to evaluate postoperative bone formation and remodeling. Rats underwent an intraperitoneal injection of tetracycline (TC) (2 mg/100 g body weight) dissolved in phosphate-buffered saline (PBS(–)) (40 mL) at 5 weeks after surgery, followed by calcein (CL) (1 mg/100 g body weight) in PBS(–) (40 mL) at 7 weeks and 7 weeks and 5 days (2 days before sacrifice) after surgery. The rat calvariae with P-nHAP/COL were then processed for non-decalcified histology. After a 1-week fixation in 70% ethanol at 4 °C, the samples were dehydrated in a graded series of ethanol (1 day at each concentration) and then placed in pure acetone for 24 h. Those samples were subsequently stained with Villanueva solution (222-01445, Wako, Osaka, Japan). Finally, the samples were embedded in methylmethacrylate for 4 days and chemically polymerized for 10 days. The non-decalcified resin blocks (~15 × 15 × 20 mm) were cut sagittally using a circular diamond cutter (MC-201 Microcutter, Maruto, Tokyo, Japan). Sections were attached to plastic slides, ground to a thickness of 20 μm using a precision lapping machine (ML-110N, Maruto, Tokyo, Japan), and then manually polished in accordance with the method of Frost [26]. As controls for comparison with P-nHAP/COL, ground sections of calvarial defects without material (DEF) were also prepared, employing identical staining and fluorescent labeling techniques. Histological observations were performed using fluorescence microscopy (All-in-one BZ-9000, Keyence, Osaka, Japan).

## 3. Results

### 3.1. Scanning Electron Microscopy (SEM) Observations

Prior to pressing the COL and nHAP/COL sponges using hydraulic press, highly inter-connected pores were in both sponges (Figure 2). The pore sizes of COL were 215.5 ± 100.9 µm and 122.7 ± 44.4 µm in short and long diameters, respectively, whereas those of nHAP/COL were 240.8 ± 70.5 µm and 93.0 ± 19.9 µm. In the nHAP/COL sponge, some aggregates of nHAP particles were observed on the COL wall, whereas most nHAP existed in COL (Figure 2b).

Characteristically, both P-COL and P-nHAP/COL were dense with plain surfaces without pores after pressing (Figure 3). In P-COL, multiple plane surfaces were observed under low magnification (Figure 3a); their micro-surfaces exhibited grooves under high magnification (Figure 3b), reflecting collagen fibers. In P-nHAP/COL, the surfaces were highly interleaved with nHAP particles under low magnification (Figure 3c), whereas agglomerated nHAP particles were found under high magnification (Figure 3d).

Thus, hydraulic press apparently altered the sponge configurations of COL and nHAP/COL (Figure 2) to bulk plain structures of P-COL and P-nHAP/COL (Figure 3). It became evident for P-nHAP/COL, nHAP powder were highly condensed in the COL matrix, compacted as aggregate, and widely exposed outside (Figure 3b,d).

### 3.2. Micro-CT

Figure 4 and Figure 5 show representative micro-CT image analysis results of the region of interest (ROI) of cranial defects and the overall X-ray opacity in the defects with and without implant materials at 3 days, 4 weeks, and 8 weeks after surgery, respectively.

With P-COL and DEF, the defects at 3 days were relatively X-ray transparent (Figure 4a,b), as neither COL nor soft tissue is X-ray opaque. The opacity of the defects with P-COL and DEF increased slightly from 3 days to 4 weeks due to soft tissue infiltration; opacity plateaued from 4 to 8 weeks (Figure 5), implying no new active bone formation. In contrast, with P-nHAP/COL, both nHAP and newly formed bone were X-ray opaque (Figure 4c). With P-nHAP/COL, the defect was originally radiopaque at 3 days, due to the radiopacity of nHAP. Although P-nHAP/COL was gradually digested, leading to reduced radiopacity in the defect zone, the radiopacity increased overall due to the ossification of new bone formation and growth. The opacity of the defect area implanted with P-nHAP/COL was significantly greater at 8 weeks after surgery than at 3 days (*p* < 0.05) (Figure 5); this reflected considerable new bone formation, which overwhelmed the reduction in radiopacity associated with degradation of P-nHAP/COL.

The results of two-way repeated-measures ANOVA (*n* = 6) of the X-ray opacity data (Figure 5) were as follows: factor of samples: *p* = 3.48 × 10^−9^ (<0.01), factor of periods: *p* = 2.33 × 10^−7^ (<0.01), and samples × periods interactions: *p* = 0.32. The defect zones implanted with P-nHAP/COL had mean opacity values of 96, 111, and 135 at 3 days, 4 weeks, and 8 weeks, respectively. P-nHAP/COL itself was moderately radiopaque in the defect at 3 days postoperatively. The opacity increased significantly from 3 days up to 8 weeks (between 3 days and 8 weeks, *p* < 0.05), indicative of active bone formation. The time-dependent bone formation was morphologically identified (Figure 4c). On the other hand, the defect zones with P-COL showed mean opacity values of 69, 84, and 90 at 3 days, 4 weeks, and 8 weeks, respectively (between 3 days and 4 weeks, *p* < 0.05; between 3 days and 8 weeks, *p* < 0.05). Those of DEF showed values of 61, 89, and 96, respectively (between 3 days and 4 weeks, *p* < 0.01; between 3 days and 8 weeks, *p* < 0.01). With both P-COL and DEF, bone formation in the defect area with time was fairly limited and sluggish (Figure 4a,b). At 3 days, 4 weeks, and 8 weeks, P-nHAP/COL always had higher X-ray opacity than that of P-COL and DEF (*p* < 0.05 or *p* < 0.01), while the opacity of the defect without implant material (DEF) was similar to that with P-COL at all time points (Figure 5). In summary, the 3D micro-CT image analyses suggested that P-nHAP/COL was osteo-conductive in rat critical-size cranial defects.

### 3.3. Histological Observations

Villanueva-stained histological images of a cranial bone defect (red rectangle) filled with P-nHAP/COL and without implant material (DEF) at 8 weeks after surgery are shown in Figure 6a,b, respectively. New bone appeared white, whereas older bone was light yellow to brown in color. When filled with P-nHAP/COL, most of the remaining fragmented P-nHAP/COL particles (indicated by *) were present in the upper half of the cranial defect. Island-like areas of new bone were detected in the lower half of the defect (indicated by NB in Figure 6a). In contrast, no active bone formation was noted in the cranial defect only (DEF), in which only connective tissues were detected (Figure 6b).

Figure 7 shows the magnified image of one spot from the red rectangle cranial bone defect area in Figure 6a, clarifying phagocytosis of fragmented P-nHAP/COL particles (*) in a cranial bone defect filled with P-nHAP/COL at higher magnification at 8 weeks after surgery. Multinucleated giant cells (virtually, osteoclasts), (#) as well as macrophages, had direct contact with and actively digested the remaining P-nHAP/COL particles (*). The nucleus appeared blue. Multinucleated giant cells (osteoclasts) (#) appeared as blue aggregates, whereas macrophages appeared as single blue dots. Collagenous tissues appeared as tangled threads. The remaining P-nHAP/COL particles (*) were brown.

Double-stained fluorescence images of a cranial bone defect filled with P-nHAP/COL and without implant material (DEF) 8 weeks after surgery are shown in Figure 8a,b, respectively. The defects filled with P-nHAP/COL showed island-like areas of new bone, localized in the lower half of the defect. Active bone formation was confirmed by double staining with yellow TC at 5 weeks after surgery and green CL at 7 weeks after surgery and at 2 days before sacrifice. Double staining clarified the dynamic bone formation activities in both inward and outward directions (Figure 8a). New bone formation occurred in one of two ways: extension of existing bone edges or island-like bone formation inside the defect zone. In contrast, no active bone formation was seen in the defect gap zone of the counterpart sample (DEF) (Figure 8b).

Figure 9 shows the magnified images of one spot from cranial defect in Figure 8a, namely, Villanueva-stained and double fluorescently labeled slices of the new bone formation zone in the cranial defect filled with P-nHAP/COL at higher magnification. In the Villanueva-stained images (Figure 9a), new bone was covered with purple (uncalcified) osteoid produced by osteoblasts, which would soon be calcified, leading to growth of bone. The new bone had many spot-like osteocytes. In TC fluorescently labeled samples (Figure 9b), a strong yellow line was observed at injection (at 5 weeks after surgery), but the yellow color diffused over 5 weeks postoperatively. In CL fluorescently labeled samples (Figure 9c), marked localization at two injection times (at 7 weeks after surgery and at 2 days before sacrifice) was reflected in the color green. In overlaid TC + CL (Figure 9d), dynamic bone formation trends were well characterized. The top two bone islands were formed quickly after 7 weeks, whereas most of the bottom bone was formed before 7 weeks.

## 4. Discussion

Our assumption discussed in the Introduction section, i.e., that the new biomaterial, P-nHAP/COL, would be osteo-conductive in rat cranial bone defects, was confirmed by both micro-CT and histological observations. P-nHAP/COL led to successful new bone formation (covering approximately 40–50% of the total area) in critical-size defects in rat skulls at 8 weeks after surgery (Figure 4c, Figure 6a and Figure 8a); this represented remarkable success. In this context, the following should be noted: bone regeneration was not achieved in the defect filled with P-COL that lacked nHAP (Figure 4b), a large single-body HAP disk sintered from nHAP particles was bio-inert and hindered bone regeneration in the defect zone in a previous study [27], the P-nHAP disk without COL in our preliminary experiments was also bio-inert and hindered bone regeneration (i.e., as a large HAP block), and the nHAP/COL porous sponge counterpart without hydraulic press was more hydrophilic with lower agglomeration energy and reduced density, compared with P-nHAP/COL, and tended to virtually disintegrate in the bone defect for up to 4 weeks after implantation [11]. Thus, effective osteo-conduction was solely facilitated by P-nHAP/COL in the rat cranial bone defect.

The success of P-nHAP/COL with respect to osteo-conduction may be attributed to two key factors, namely the use of (1) hydraulic press and (2) nHAP. (1) Application of hydraulic press considerably changed the physical state of nHAP/COL from a soft porous sponge to a highly condensed hard disk. With intensified physical agglomeration energy (i.e., surface energy), the P-nHAP/COL composite disk became more hydrophobic and less bio-absorbable [12], allowing bone regeneration in rat critical-size cranial bone defects for the period up to 8 weeks. (2) HAP in HAP/COL composites showed different bone regeneration efficacy depending on size (i.e., macro-, micro-, or nano-size) [28]. The nano-size of nHAP in the composite seemed to play an important role in rapid bone regeneration [29]. Precursors of multinucleated giant cells (i.e., osteoclasts) attached to, differentiated, and matured on P-nHAP/COL (Figure 7). Low-level dissolution of calcium and phosphate ions from nHAP in P-nHAP/COL might be chemotactic to precursor cells [30], followed by differentiation and maturation of osteoclasts. COL is minimally osteo-conductive, partly because it serves as a scaffold for stem cells and osteoblasts [31], and is also an excellent physical stabilizer for P-nHAP. Indeed, natural bone consists largely of COL and nHAP [32], and its basic structure resembles that of our P-nHAP/COL sample [33]. The COL portion of the composite may assist in formation of the ruffled border of mature osteoclasts [34]. Proton pump H^+^-adenosine triphosphatase (ATPase) of mature osteoclasts could secrete H^+^ ions, which effectively dissolve nHAP, while the enzyme Cathepsin K from osteoclasts efficiently degrades highly condensed P-COL [35]. Dual use of hydraulic press and n-HAP is recommended in future biomaterial applications to prepare new HAP/COL-based biomaterials. In addition, growth factors could be coupled with n-HAP for accelerated bone regeneration [36].

Rapid bone regeneration at the rat cranial bone defect was apparently achieved by bone remodeling, in which the presence of multinucleated giant cells (osteoclasts) was important (Figure 7). Notably, new bone formation occurred in the zones beneath or near fragmented P-nHAP/COL particles, in contact with and digested by multinucleated giant cells (osteoclasts) (Figure 6a, Figure 7 and Figure 8a). This bone remodeling [37] may be essential for rapid bone regeneration within critical-size rat cranial defects [38]. As mentioned, disintegrating nHAP/COL particles may attract precursor cells, which then differentiate and mature into osteoclasts [39], leading to rapid bone remodeling. Osteoclasts also produce soluble factors (e.g., receptor activator of nuclear factor κB (RANK)), stimulating and activating nearby pre-osteoblasts and osteoblasts expressing RANK ligand (RANKL) [40]. As a result, a bone remodeling system mediated by osteoblasts and osteoclasts [37,40,41] may be established in rat cranial bone defects filled with P-nHAP/COL, thus facilitating active bone remodeling. P-nHAP/COL appeared to act as old natural bone to be replaced [42]. Dissolution of calcium and phosphate ions from nHAP may further cause angiogenesis in the defect zone [43], thus aiding bone formation.

The P-nHAP/COL applied in implant surgery can be in disk, particulate, or film form. Before P-nHAP/COL is applied clinically in humans, successful treatment of cranial bone defects must be demonstrated in studies using larger animals, such as dogs and monkeys, because bone-forming capability in cranial areas varies among animal species; bone defects of rats are regenerated more quickly compared with larger animals and humans [44].

The overall findings of this study are as follows: P-nHAP/COL was highly osteo-conductive in critical-size calvarial defects in rats, promoting the formation of newly regenerated bone (covering up to 50% of the total area) for up to 8 weeks after implantation. Bio-degradation of P-nHAP/COL effectively facilitated osteo-conduction via bone-remodeling. P-nHAP/COL could further increase bone regeneration over 8 weeks (to >50% of the total cranial defect area) and might assist in bone regeneration in dental patients with an atrophic alveolar ridge in the near future.

Referring to current alloplasts employed in clinical dentistry, one commercial sintered apatite bone substitute material (Spongious granules 0.25 to 1 mm) (Bio-Oss, Geistlich Pharma AG, Wolhusen, Switzerland) was preliminary tested in the rat cranial bone defect. This material is considered as deproteinized bovine bone mineral, and has been most utilized worldwide [45]. However, this material was very inert and did not exert effective osteo-conductive activities in rat cranial bone defects at 8 weeks after surgery (Appendix A). In clinical situation, a longer period exceeding 6 to 7 months was required to bio-absorb this material, followed by new bone formation [46]. Thus, P-nHAP/COL could be more suitable for enabling quick and large-volume bone formation in bone defect areas without use of growth factors [36].

## 5. Conclusions

Within the limitations of this study, the evaluation of P-nHAP/COL (pressed nano-apatite/collagen) composite can be summarized as follows:P-nHAP/COL composite was prepared by mechanical mixing, freeze drying, dehydrothermal cross-linking, and hydraulic press. The composite was punched into critical-size disks.P-nHAP/COL disks were implanted into rat critical-size cranial defects, and bone regeneration status was evaluated by micro-CT imaging and histological examination. After 8 weeks of observation, P-nHAP/COL was highly osteo-conductive in vivo. The defect zone implanted with P-nHAP/COL rapidly became X-ray opaque, indicating newly formed bone. Active bone remodeling composed of both osteoblasts and osteoclasts was histologically observed.P-nHAP/COL could be used as a new osteo-conductive bone substitute in dental implants, although further studies using larger animals (e.g., rabbits and dogs) are needed prior to clinical testing.P-nHAP/COL would be applied to bone-shallow area, followed by dental implant treatment (Figure 10).

## Figures and Tables

**Figure 1 materials-15-03376-f001:**
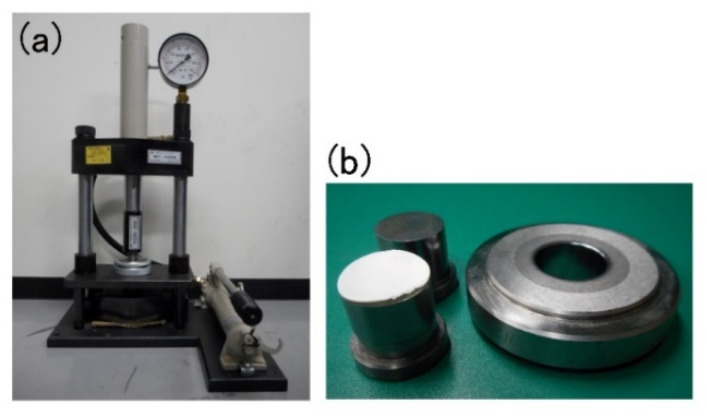
(**a**) A hydraulic press machine used in this study, (**b**) specimen on the die.

**Figure 2 materials-15-03376-f002:**
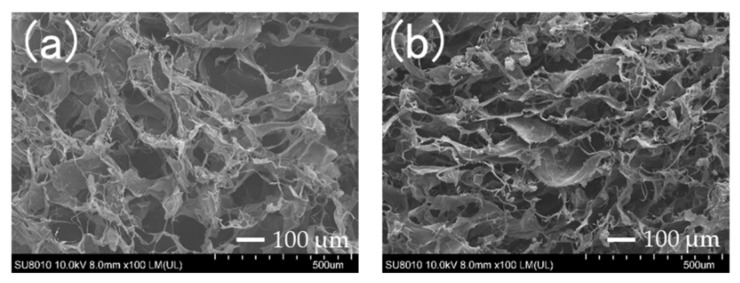
Scanning Electron Microscopy (SEM) photographs of sponges ((**a**): collagen (COL) and (**b**): nano-hydroxyapatite/collagen (nHAP/COL)) prior to hydraulic press at low magnification (×100).

**Figure 3 materials-15-03376-f003:**
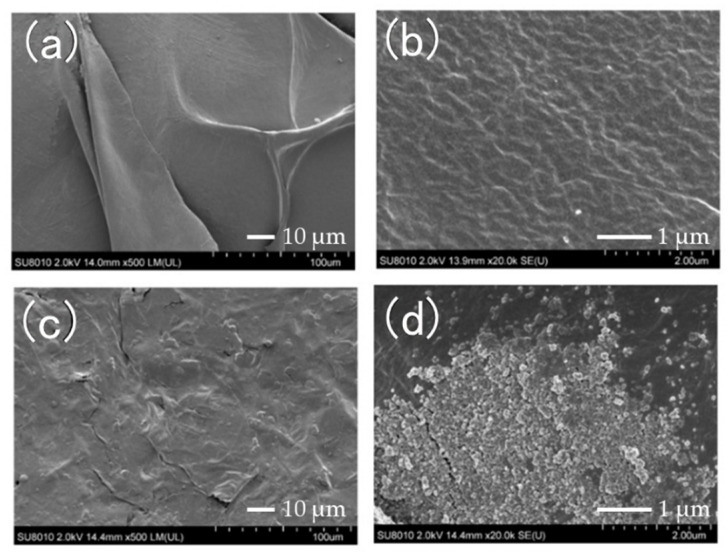
SEM photographs of pressed collagen (P-COL) and pressed nano-hydroxyapatite/collagen (P-nHAP/COL) ((**a**): P-COL at low magnification (×500), (**b**): P-COL at high magnification (×20,000), (**c**): P-nHAP/COL at low magnification (×500), and (**d**): P-nHAP/COL at high magnification (×20,000)).

**Figure 4 materials-15-03376-f004:**
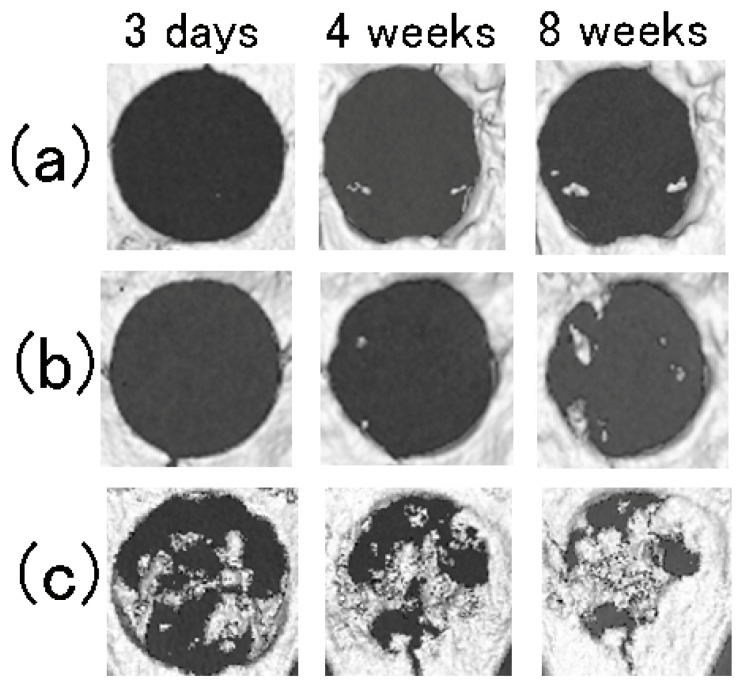
Micro-computed tomography (CT) images of cranial defects with and without implant materials ((**a**): defect only (DEF), (**b**): P-COL, and (**c**): P-nHAP/COL)) at 3 days, 4 weeks, and 8 weeks after surgery.

**Figure 5 materials-15-03376-f005:**
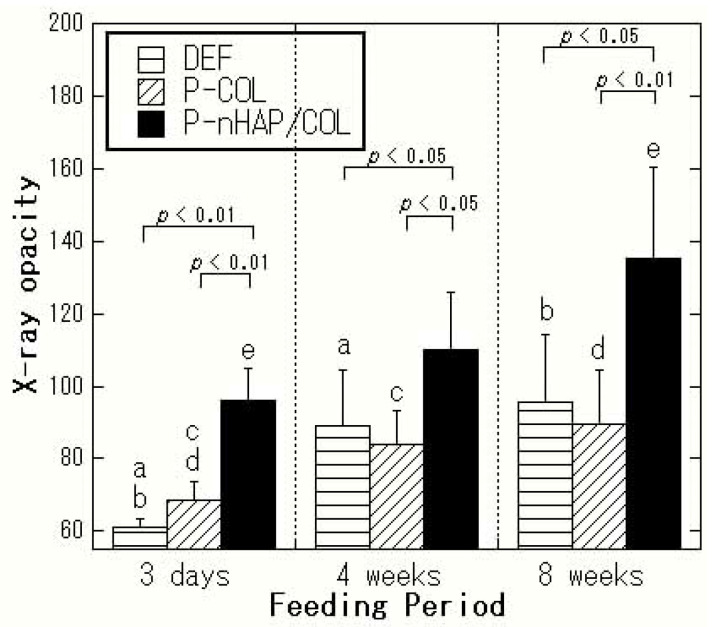
Graphs of X-ray opacity in the defects with and without implant materials (DEF, P-COL, and P-nHAP/COL) (*n* = 6). Note: Pairs indicated with the same letter were significantly different. Note: a and b, *p* < 0.01; c–e, *p* < 0.05.

**Figure 6 materials-15-03376-f006:**
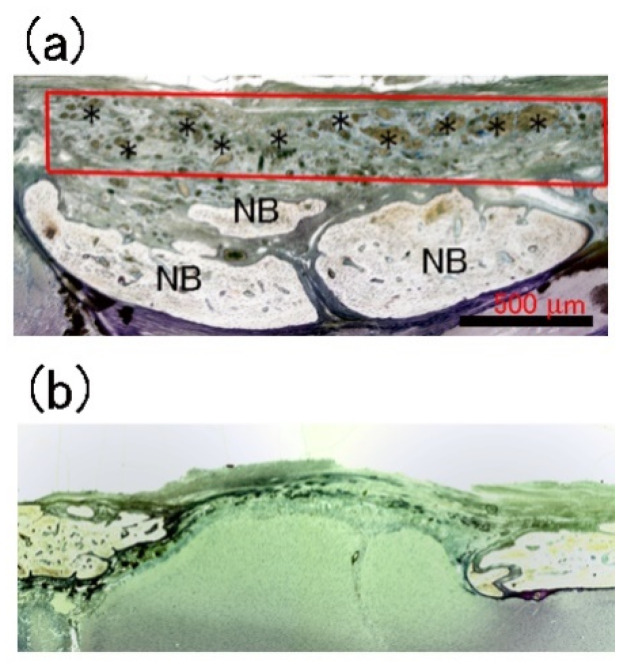
Villanueva-stained histological images of cranial bone defect filled with P-nHAP/COL ((**a**): red rectangle) and without implant material (DEF: (**b**)) at 8 weeks after surgery. Note: (**a**) Most fragmented P-nHAP/COL remnants (*) existed in the upper half of the cranial defect (red rectangle). NB = newly formed bone. (**b**) Connective tissues alone were seen in the defect gap.

**Figure 7 materials-15-03376-f007:**
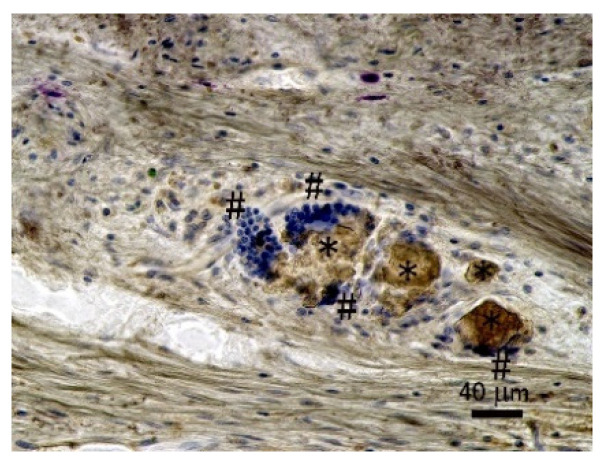
Magnified image of one spot from red rectangle cranial bone defect area in Figure 6a. Note: Nuclei were stained blue. Multinucleated giant cells (osteoclasts) (#) and single-nucleated macrophages were in contact with and digested fragmented remnant P-nHAP/COL particles (*).

**Figure 8 materials-15-03376-f008:**
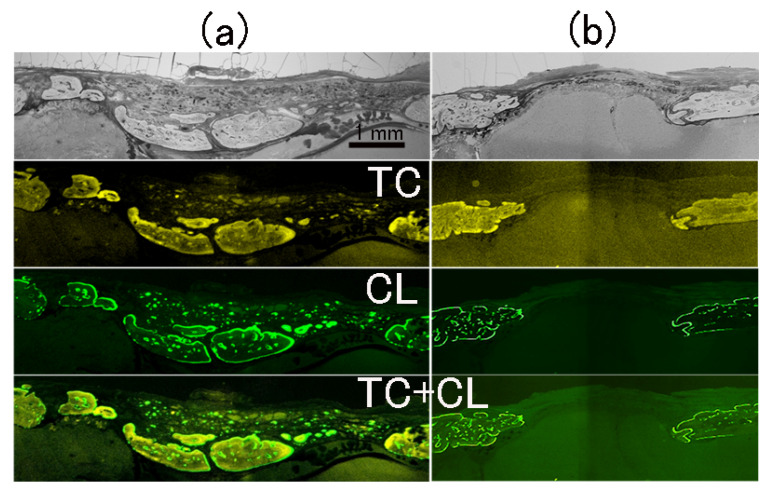
Double fluorescently stained images of cranial bone defect filled with P-nHAP/COL (**a**) and without implant material (DEF; (**b**)) at 8 weeks after surgery. Top = contrast; second = TC, fluorescently labeled; third = CL, fluorescently labeled; bottom = TC + CL overlay images.

**Figure 9 materials-15-03376-f009:**
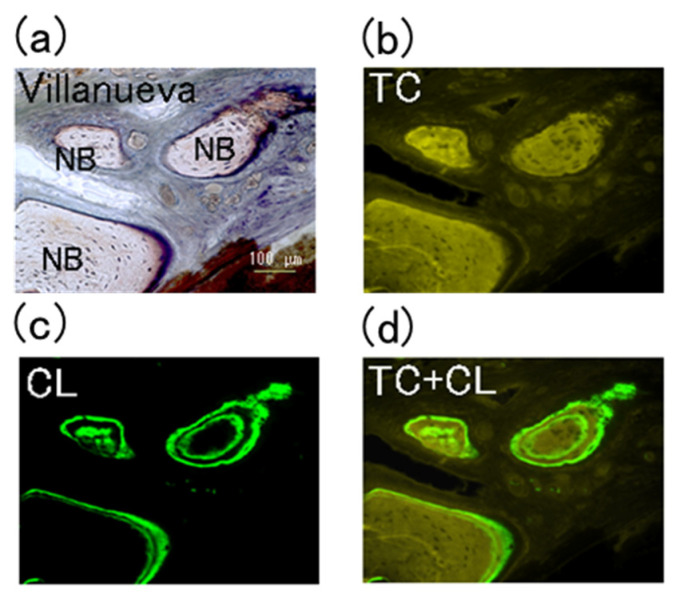
The magnified images one spot from cranial bone defect area in Figure 8a, namely, Villanueva-stained histological image of the new bone formation zone obtained from the cranial bone defect filled with P-nHAP/COL at 8 weeks after surgery at higher magnification (**a**) and double fluorescently labeled images such as TC fluorescent labeling (**b**). CL fluorescent labeling (**c**), TC + CL overlay (**d**). Note: In Villanueva-stained images (**a**), new bones (NB) with osteocytes (dots) were lined with purple-stained osteoid.

**Figure 10 materials-15-03376-f010:**
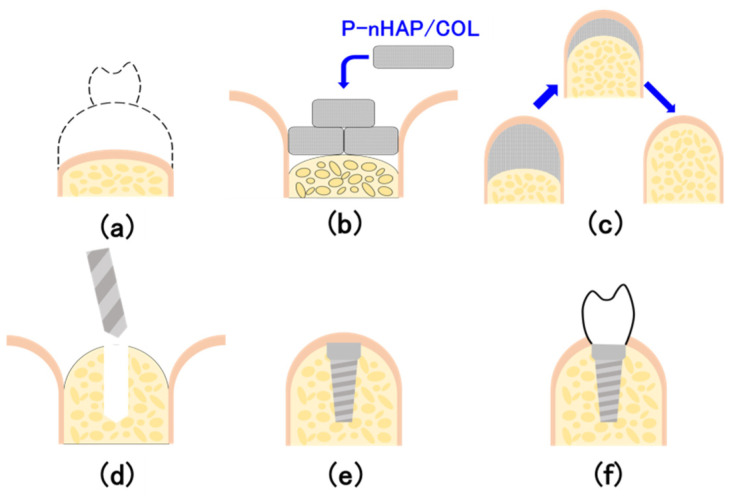
Schematic illustration of the future use of P-nHAP/COL: (**a**) bone-shallow area on mandible, (**b**) material application, (**c**) bone augmentation, (**d**) drilling and tapping for dental implant, (**e**) implant insertion, bone healing, and osteo-integration, and (**f**) screwing super-structure to dental implant.

## Data Availability

All data are included in the manuscript.

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
