# Peer review of "Bone Regeneration of Critical-Size Calvarial Defects in Rats Using Highly Pressed Nano-Apatite/Collagen Composites"

_materials, 2022, doi:10.3390/ma15093376_

Round 1

Reviewer 1 Report

The authors submitted a research article with the aim of elucidating the potencies of highly pressed nano-hydroxyapatite/collagen (P-nHAP/COL) composites in connection with osteo-conductive bone substitution. The authors evaluated  their utility in a rat-skull critical-size bone-defect model using micro-CT analyses and histological observations. Animal model was constructed from Wistar rats. The author found that P-nHAP/COL was strongly osteo-conductive and could serve as a useful novel bone substitute material for future dental implant treatments. The aim of the study is clear and concise. The manuscript is well-balanced and logically structured. It has clear tables and legible figures and did not duplicate the information and findings incorporated into the whole text of the paper. The methods are well described and thoroughly referenced. Section Discussion covers all aspects of the study and proposes to the redears to clearly understand the role of novel materials in the dental implant strategy. Overall, this is excellent work and I congratulate the authors on it. However, I would like to put forward some questions to discuss.

  1. The authors seem to extend the section Conclusion to clearly illustrate the clinical significance of the findings.
  2. In addition to that, some comparissons of current and conventional materials in the dental implants could improve the impression of the study

Author Response

・General comment:

We would like to thank the reviewer for careful and thorough reading of our manuscript.

According to your kind suggestion, we have corrected the manuscript. Corrected passages have been yellow highlighted in the whole of manuscript.

  1. The authors seem to extend the section Conclusion to clearly illustrate the clinical significance of the findings.

Answer: We have added the description of the summary and the illustration (Figure 10) (Page 11, lines 399-406).

  1. P-nHAP/COL would be applied to bone-shallow area, followed by dental implant treatment. (Figure 10).

Figure 10. Schematic illustration of the future use of P-nHAP/COL; (a) bone shallow area on mandible, (b) material application, (c) bone augmentation, (d) drilling and tapping for dental implant, (e) implant insertion, bone healing and osteo-integration, and (f) screwing super-structure to dental implant.

  1. In addition to that, some comparisons of current and conventional materials in the dental implants could improve the impression of the study

Answer: We have added the descriptions and the figure (Figure S1) of conventional alloplasts and compared with our results (Page 10, lines 367-383). We have also added the references (No. 45-46) (Page 13, line 517-page 14, line 520).

Referring to current alloplasts employed in clinical dentistry, one commercial sintered apatite bone substitute material (Spongious granules 0.25 to 1 mm) (Bio-Oss, Geistlich Pharma AG, Wolhusen, Switzerland) was preliminary tested in the rat cranial bone defect. This material is considered as deproteinized bovine bone mineral, and has been most utilized worldwide [45]. However, this material was very inert and did not exert effective osteo-conductive activities in rat cranial bone defect at 8 weeks after surgery (Figure S1). In clinical situation, longer period exceeding 6 to 7 months was required to bio-absorb this material, followed by new bone formation [46]. Thus, P-nHAP/COL could be more suitable for enabling quick and large-volume bone formation in bone defect areas without use of growth factors [36].

Figure S1. Villanueva-stained histological images of the zone obtained from the rat cranial bone defect filled with one commercial alloplast (Bio-Oss) at 8 weeks after surgery at low magnification (top); that at high magnification (middle-left) with CL- fluorescent labeling (middle-right); micro-CT sagittal section image (bottom). Note: Bio-degradation of the material was minimum in cranial defect, and multinucleated-giant cells (osteoclast) was not observed. Osteoid formation preceded by CL labeling was very sluggish.

Reviewer 2 Report

Collagen/Hap composite materials have been used widely as scaffolds for tissue engineering applications. The present manuscript is a continuation of  previous articles from the same group [references 11, 13]. The basic difference is that they pressed their scaffolds in an hydraulic press and they did the experiments in vivo.

The authors did in vivo experiments and a detailed histological and structural characterization (micro-CT) of their materials and found that using pressing their material has better osteoconductive properties.

My only concern is the novelty, however since these “pressed” scaffolds have not been used previously in vivo in animal models my suggestion is that the manuscript can be marginally accepted in “Materials” after major revision.

Comments

  1. Introduction part  is too short
  2. In the paper of the same group [Reference 13] the authors claim that “We produced highly pressed nano-hydroxyapatite/collagen composites (P-nHAP/COL) by Newton press. Why the material in the submitted manuscript is presented as novel?
  3. Give more details and discuss about the  scaffold’s architecture  including pore size and size distributions,
  4. I would suggest using the term “hydraulic press” instead of “Newton press”

Author Response

・General comment:

Thank you for your thorough review and your helpful advice and suggestions. We carefully considered your comments. Corrected passages have been yellow highlighted in the whole of manuscript.

  1. Introduction part is too short.

Answer: We have added and rewritten the sentences to clarify the finding of our previous study and the aim of this study (Page 2, lines 52-79). Also, we have added the references (No. 13-17, 19) (Page 12, line 450- page 13, line 461, and page 13, lines 464-465).

In our previous in vivo study, nHAP/COL composite (soft porous sponge without pressing) was prepared by mechanical mixing and freeze-drying [11]. This scaffold material indicated osteo-conductive at 4 weeks in rat calvarial bone defects, however, new bone formation at 8 weeks was slightly decreased due to rapid bio-absorption of the material.

Thus, further improvement was needed for achieving a prolonged osteo-conductive effect of nHAP/COL composite. One possible method to address this requirement was the condensation of composite using a hydraulic press [12]. This technique has already employed to decellularize and sterilize animal skin and meat in food industry [13,14]. Under pressing, collagen structure (e.g., alpha-helix and beta-sheet) is significantly altered, changing the functional and structural properties of collagen, dependent on the magnitude and period of pressure [13,14]. Pressed collagen might possess increased hydrophobicity, compressive strength and in vivo longevity [13]. On the other hand, for preparation of HAP, cold isostatic pressing was used as one process for raw materials prior to sintering [15]. Although once sintered HAP might not be chemically altered by pressing, pressing might cause HAP particles to agglomerate, leading to different physical properties [16]. Also, hot isostatic pressing has often been employed to produce dense and highly crystalline HAP [17]. In our previous studies, the pressing technique could embed HAP particles within COL with greater physical energy and the osteogenic differentiation of osteoblasts was accelerated on pressed nHAP/COL (P-nHAP/COL) composite compared with pressed COL(P-COL) in vitro [12,18]. However, the evidence of P-nHAP/COL is still insufficient for clinical use.

Up to now, there have been little reports concerning the effect of high pressure on not only physical and chemical properties but also in vivo osteo-conduction of P-nHAP/COL composites. It is highly expected to unveil these properties. Current most bulk scaffold materials are porous with interconnected porosity [19]. P-nHAP/COL composites are, however, plain and dense without pore structure, and their usefulness remains unknown. The novelty of this research lied in their clarification.

References:

  1. Gao, Y.; Wang, L.; Qiu, Y.; Fan, X.; Zhang, L.; Yu, Q. Valorization of cattle slaughtering industry by-products: modification of the functional properties and structural characteristics of cowhide gelatin induced by high hydrostatic pressure. Gels. 2022, 8(4), 243.
  2. Chen, L.; Ma, L.; Zhou, M.; Liu, Y.; Zhang, Y. Effects of pressure on gelatinization of collagen and properties of extracted gelatins. Food. Hydrocoll. 2014, 36, 316–322.
  3. Itoh, H.; Wakisaka, Y.; Ohmura, Y.; Kuboki, Y. A new porous hydroxyapatite ceramic prepared by cold isostatic pressing and sintering synthesized flaky powder. Dent. Mater. J. 1994, 13(1), 25-35.
  4. Kalita, S.J.; Bose, S.; Hosick, H.L.; Bandyopadhyay, A. CaO--P2O5--Na2O-based sintering additives for hydroxyapatite (HAp) ceramics. Biomaterials. 2004, 25(12), 2331-2339.
  5. Buchi Suresh, M.; Biswas, P.; Mahender, V.; Johnson, R. Comparative evaluation of electrical conductivity of hydroxyapatite ceramics densified through ramp and hold, spark plasma and post sinter hot isostatic pressing routes. Mater. Sci. Eng. C. Mater. Biol. Appl. 2017, 70(pt 1), 364-370.
  6. Qin, D.; Wang, N.; You, X-G.; Zhang, A-D.; Chen, X-G.; Liu, Y. Collagen-based biocomposites inspired by bone hierarchical structures for advanced bone regeneration: ongoing research and perspectives. Biomater. Sci. 2022, 10(2), 318-353.

  1. In the paper of the same group [Reference 13] the authors claim that “We produced highly pressed nano-hydroxyapatite/collagen composites (P-nHAP/COL) by Newton press. Why the material in the submitted manuscript is presented as novel?

Answer: Please see the answer 1. We have added and rewritten the sentences to clarify the finding of our previous study and the aim of this study (Page 2, lines 52-79).

  1. Give more details and discuss about the scaffold’s architecture including pore size and size distributions.

Answer: Thank you for your suggestion. We have added the descriptions about the scaffold’s architecture including pore size and size distributions

Materials and Methods:

  • We have added and corrected the descriptions of SEM observations to add the analysis of the P-COL and P-nHAP/COL sponges before pressing (Page 3, lines 111-116).

      Scanning electron microscopy (SEM) observations of both COL and nHAP/COL before and after hydraulic press were performed by using a scanning electron microscope (SU8010, Hitachi High-Tech Corp., Tokyo, Japan) at 2 or 10 kV after plasma-coating with OsO4 to understand morphological changes by hydraulic press. In addition, the pore sizes of COL and nHAP/COL sponges (pre-forms of P-COL and P-nHAP/COL) were calculated from different locations (n = 20) per one sample.

Results:

  • We have added the analysis of SEM observations of the COL and nHAP/COL sponges before pressing and calculation of thier pore size (Page 4, lines 176-181). Also, we have added the figure (Figure 2) (Page 5, lines 192-194).
  • We have added the descriptions about the comparisons of the samples before and after pressing (Page 4, line 188-page 5, line 191).

Prior to pressing the COL and nHAP/COL sponges using hydraulic press, highly inter-connected pores were in both sponges (Figure 2). The pore sizes of COL were 215.5 ± 100.9 µm and 122.7 ± 44.4 µm in short and long diameters, respectively, whereas those of nHAP/COL were 240.8 ± 70.5 µm and 93.0 ± 19.9 µm. In nHAP/COL sponge, some aggregates of nHAP particles were observed on the COL wall, whereas most nHAP existed in COL (Figure 2b).

Thus, hydraulic press apparently altered the sponge configurations of COL and nHAP/COL (Figure 2) to bulk plain structures of P-COL and P-nHAP/COL (Figure 3). It became evident for P-nHAP/COL, nHAP powder were highly condensed in COL matrix, compacted as aggregate, and widely exposed outside (Figure 3b and d).

Figure 2. SEM photographs of sponges (a: COL and b: nHAP/COL) prior to hydraulic press at low magnification (×100).

  1. I would suggest using the term “hydraulic press” instead of “Newton press”

Answer: Thank you for your suggestion. We have corrected the term “Newton press” to “hydraulic press” in the whole of the manuscript.

Round 2

Reviewer 2 Report

The revised manuscript is improved and can be accepted for publication